# Distribution of antibiotic resistance genes and antibiotic residues in drinking water production facilities: Links to bacterial community

**Karabo Tsholo**[1]*, **Lesego Gertrude Molale-Tom**[1], **Suranie Horn**[1,2], **Cornelius Carlos Bezuidenhout**[1]

**1** Unit for Environmental Sciences and Management – Microbiology, North-West University, Potchefstroom, South Africa, **2** Occupational Hygiene and Health Research Initiative (OHHRI), Faculty of Health Science, North-West University, Private Bag X6001, Potchefstroom, South Africa

* 24245968@g.nwu.ac.za

**Data Availability Statement:** All relevant data are within the manuscript and its Supporting information files.

## Abstract

There is a rapid spread of antibiotic resistance in the environment. However, the impact of antibiotic resistance in drinking water is relatively underexplored. Thus, this study aimed to quantify antibiotic resistance genes (ARGs) and antibiotic residues in two drinking water production facilities (NW-E and NW-C) in North West Province, South Africa and link these parameters to bacterial communities. Physicochemical and ARG levels were determined using standard procedures. Residues (antibiotics and fluconazole) and ARGs were quantified using ultra-high performance liquid chromatography (UHPLC) chemical analysis and real-time PCR, respectively. Bacterial community compositions were determined by high-throughput 16S rRNA sequencing. Data were analysed using redundancy analysis and pairwise correlation. Although some physicochemical levels were higher in treated than in raw water, drinking water in NW-E and NW-C was safe for human consumption using the South African Water Quality Guideline (SAWQG). ARGs were detected in raw and treated water. In NW-E, the concentrations of ARGs (*sul1*, *intl1*, EBC, FOX, ACC and DHA) were higher in treated water than in raw water. Regarding antimicrobial agents, antibiotic and fluconazole concentrations were higher in raw than in treated water. However, in NW-C, trimethoprim concentrations were higher in raw than in treated water. Redundancy analysis showed that bacterial communities were not significantly correlated (Monte Carlo simulations, p-value >0.05) with environmental factors. However, pairwise correlation showed significant differences (p-value <0.05) for *Armatimonas*, *CL500-29 marine group*, *Clade III*, *Dickeya* and *Zymomonas* genera with environmental factors. The presence of ARGs and antibiotic residues in the current study indicated that antibiotic resistance is not only a clinical phenomenon but also in environmental settings, particularly in drinking water niches. Consumption of NW-E and NW-C treated water may facilitate the spread of antibiotic resistance among consumers. Thus, regulating and monitoring ARGs and antibiotic residues in drinking water production facilities should be regarded as paramount.

**Funding:** This work is based on the research supported in part by the National Research Foundation of South Africa Grant No. C2019-2020-00224 (Bursary for Karabo Tsholo), The Water Research Commission (WRC) of South Africa: (Contract - 2019/2020-00224). The views expressed are those of the authors and not of the funding agencies.

**Competing interests:** The authors have declared that no competing interests exist.

## Introduction

Antibiotics have revolutionized medicine and improved the prevention and treatment of infectious bacteria, thus reducing the mortality burden on humans and animals. Hence, antibiotic use has become common worldwide, leading to the accumulation and spread of antibiotic resistance [1, 2]. In addition, the lack of development of new generations of antibiotics has threatened the effective treatment of infectious bacteria with common antibiotics [3]. Unfortunately, antibiotic resistance complicates treatment requiring extended hospitalization and high-cost drugs to treat common bacterial infections [4, 5].

Antibiotic resistance may be high in South Africa since the country has a heightened usage of antibiotics comparable and sometimes higher than those in BRICS (Brazil, Russia, India, Canada and South Africa) nations, the United States and the United Kingdom [6]. The aforementioned report further revealed that in 2015, more than 60% of the antibiotics procured in South Africa for human health were used as growth promoters. In addition to this high use of antibiotics, the antifungal, fluconazole, is prescribed to people living with HIV/AIDS (PLWH) and since South Africa has 13.7% PLWH, fluconazole is highly used [7, 8] and was included in the analysis. The extensive usage of antimicrobials has led to the spread of antibiotic residues in the aquatic environment since human and animal bodies cannot fully metabolize antibiotics [9].

Aquatic environments are important vectors of antibiotic resistance, which provide favourable conditions for bacterial growth and spread antibiotic resistance genes (ARGs) via horizontal gene transfer to other species [10, 11]. However, in South Africa, antibiotic resistance is not monitored and regulated in aquatic environments despite being reported in drinking water [12, 13]. Furthermore, studies addressing antibiotic resistance in drinking water focus on detecting ARGs and not so much on quantifying ARGs and antibiotic residues. Antibiotic and ARG concentrations are essential for developing quantitative risk assessments on consumers [14, 15].

Moreover, most studies resorted to culture-dependent techniques due to the complexities of sampling drinking water for culture-independent techniques [16]. Culture-dependent techniques do not account for viable but non-culturable bacteria that may be opportunistic bacteria harbouring ARGs [17]. High-throughput 16S rRNA sequencing could be used to overcome this limitation [18, 19]. Thus, this study aimed to quantify antibiotic resistance genes and antimicrobial residues in two drinking water production facilities (NW-E and NW-C) in North West Province, South Africa and to understand how bacterial community composition is linked to ARGs, antimicrobial residues and physicochemical parameters.

## Materials and methods

### Study area

Raw and treated water samples were collected in two DWPFs (NW-E and NW-C) in North West, South Africa. The NW-E DWPF has a total design capacity of 736 Mℓ/day and relies on two raw water sources to produce drinking water [20]. This plant services a population of 162 762 [21]. Treatment processes involved in NW-E DWPF include coagulation-flocculation, sedimentation, sand filtration, activated carbon and chlorination. The NW-C DWPF, with a total design capacity of 14 Mℓ/day, use physical separation, sand filtration and chlorination to treat water for the population of 28 720 [20, 22]. The municipality permitted the study of these DWPFs on a condition of anonymity.

## Sampling

Grab samples were collected in sterile 1 L bottles (Duran Schott, Germany) to isolate environmental DNA (eDNA). Acid-washed (free from chemical contaminants) 1 L bottles (Duran Schott, Germany) were used for the collection of samples for chemical analysis. However, up to 5 L of samples used for isolation eDNA were collected due to the complexities of sampling treated water [16]. Sampling was from October 2020 until March 2022. Samples were collected and transported in ice to the North-West University laboratories, where water analyses commenced within 24 hours of sampling.

*In situ*, total dissolved solids (TDS), salinity, temperature and pH were measured using a portable multimeter (PCSTestr 35, Eutech Instruments Pte Ltd. Singapore). Hach Lange DR 2800 spectrophotometer (HACH, USA) was used to measure nitrite (HACH method 8153), nitrate (HACH method 8039), phosphorous (HACH method 8178) and chemical oxygen demand (COD; HACH method 8000) according to the Manufacturer's protocol. Physico-chemical levels were compared to [23] for domestic use to determine the water quality of treated water.

## DNA isolation

The environmental DNA was extracted from cellulose nitrate filters with 0.2 μm porous size (Sartorius, Germany) using the Powerwater kit (Qiagen, Netherlands) per the Manufacturer's protocol. DNA concentrations were measured using the NanoDrop-1000 spectrophotometer (Nanodrop Technologies, USA) and where applicable, the DNA concentrations were diluted to 20 ng/μL for downstream applications.

## Library preparation of next-generation sequencing for 16S rRNA amplicon libraries and sequencing

The library preparation and sequencing were done according to the Illumina (USA) manufacturer's protocol. The locus-specific primers were linked to Illumina forward and reverse adapter overhang nucleotides sequence. The Agencourt AMPure XP beads (Beckman Coulter Genomics, USA) were used to purify amplicons after each PCR stage. The amplified PCR products were quantified by a Qubit 3.0 fluorometer (ThermoFisher, USA). The 16S rRNA libraries were pooled, denatured (0.2 N NaOH) and pair-end sequenced (2×300 bp) on the Illumina Miseq platform (Illumina, USA).

## 16S rRNA GENE metabarcoding analysis

Raw data sequence files were exported to QIIME2 version 2022.2 to de-multiplex, remove poor-quality sequences, and trim adaptor and primer sequences using the DADA2 pipeline. Furthermore, the DADA2 pipeline was used for clustering and taxonomic classification of operational taxonomic units (OUTs) with a 97% identity threshold—the SILVA 132 database to align the sequences. The sequences were then normalized and sequences read ranged between 47 911 and 86 187 counts per sample.

Microorganisms belonging to domain bacteria were exported to Microsoft Excel and MicrobiomeAnalyst for downstream analyses [24]. Microsoft Excel was used to construct stacked columns for bacterial phyla and genera. MicrobiomeAnalyst was used to calculate the alpha diversity based on the Shannon index (species diversity and evenness), Simpson index (species dominance) and Chao1 and ACE estimator (richness) [24].

## Screening of antibiotic resistance genes

The eDNA was subjected to the end-point PCR for screening the presence of ARGs such as *ermF*, *ermB*, *sul1*, *sul2*, *intI1*, *ampC Bla_{TEM}* and plasmid-mediated *AmpC* β-lactamase genes (pAmpCs including DHA, MOX, FOX, CIT, ACC and EBC). Each gene had a PCR reaction mixture consisting of 9.5 µl nuclease-free water (Fermentas Life Sciences, US), 12.5 µL Dream-Taq PCR master mix (Thermo Scientific, US), 1 µL of each oligonucleotide primer and 1 µL of DNA template. Furthermore, a no template control of each gene was also included. The oligonucleotide primers, amplification size and PCR cycling conditions are shown in S1 and S2 Tables.

## Quantification of 16s rRNA and antibiotic resistance genes

The absolute quantification of ARGs was conducted using the method described by [25] for pAmpCs, and [26] for 16S rRNA, *sul1* and *intl1* genes. Each gene consisted of a 10 µL reaction mixture volume in accordance with the PowerUp SYBR™ Green Master Mix manufacturer's protocol (Applied Biosystems, Thermo Fisher Scientific, Lithuania) by using the QuantStudio™ 3 platform (Applied Biosystems, Thermo Fisher Scientific, USA). The 16S rRNA, *sul1* and *intl1* genes oligonucleotide primers, amplification size and PCR cycling conditions are shown in S3 Table. The FAM fluorescent dye TaqMan gene expression assays (Thermo Fisher Scientific, USA) were used for the quantification of DHA (Ba04646120_s1), FOX (Ba04646126_s1), ACC (Ba04646144_s1), CIT (Ba04646135_s1), MOX (Ba04646156_s1) and EBC genes (Ba04646124_s1) using TaqMan's default cycling conditions.

## Solid-phase extraction and ultra-performance liquid chromatography

A quantity of 1 g of Na$_2$EDTA was added to 1 L of the samples before the extraction and the pH was adjusted to 2 with 32% HCl (Rochelle Chemicals, RSA). The HLB-H disks (Atlantic disk, USA) and an automated SPE-DEX system (Horizon Technology, Salem, NH, USA) were used for the extraction of antibiotics and fluconazole. The HLB-H disks were conditioned with methanol, ultrapure water and ultrapure water of pH 2. Samples were washed with ultrapure water and rinsed with methanol and Acetone/Methanol (1:1). The elutes were evaporated to dryness under a gentle stream of nitrogen gas within a water bath of 37˚C and the residues were dissolved in 1 ml of methanol- Milli-Q water (1:1) and 0.1% formic acid.

The analyses of antibiotics and fluconazole were performed on ultra-high-performance liquid chromatography coupled to a quadrupole time-of-flight mass spectrometry (UHPLC-Q-TOF/MS) system. Antibiotic and fluconazole were separated using a Poroshell Bonus-RP 120 (2.1×100mm; 2.7 µm) column (Agilent Technologies, USA). The mobile phase consisted of 0.1% formic acid in Milli-Q water (A) and 0.1 formic acid in acetonitrile (B). The gradient elution was 0–2.0 min, 5% B with 0.2 mL/min flow rate; 2.0–5.55 min, 15% B with 0.4 mL/min flow rate; 5.55–11.10 min, 20% B with 0.2 mL/min flow rate; 11.10–13.15 min, 25% B with 0.6 mL/min flow rate; 13.15–17.20 min, 35% B with 0.2 mL/min flow rate; 17.20–20.0 min, 100% B with 0.2 mL/min flow rate; 20.0–22.0 min, 5% B with 0.2 mL/min flow rate. The mass spectrometry parameters were drying gas temperature 250ºC, drying gas flow 8 L/min, nebulizer pressure 20 psig, sheath gas flow 7.5 L/min, VCap 3 000 V, nozzle voltage 0 V, fragmentor 130 V, skimmer 70 V and Oct 1 RF Vpp 750 V with positive electrospray ionization.

## Quality control and analytical performance

The method for antibiotic and fluconazole was validated by determining the matrix effects, linearity, limits of quantification (LOQ), limits of detection (LOD), accuracy and precision [27].

The linearity of a matrix-matched calibration curve was verified by spiking the matrix with known concentrations of antibiotics and fluconazole (ranging from 0 to 10 μg/L, depending on the target compound). The following equations calculated the LOQ and LOD:

$$LOD = {3 * Sd}/{m}$$

$$LOQ = {10 * Sd}/{m}$$

Where: Sd = standard deviation

 m = slope

The measurements for inter-day precision were performed by calculating the ratio of the standard deviation and average of the six replicates across three days. The intra-day precision was also determined [27]. The method's accuracy was determined by spiking a matrix with known low, medium and high fluconazole and antibiotic concentrations to determine the recovery. The linear equation (y = mx + c) was used to calculate the concentrations to determine the abundance ratio. Furthermore, this equation was to determine unknown concentrations of water samples. Measurements of samples were performed in triplicates.

### Risk quotient

Data generated from the chemical analysis was used to calculate the risk quotient (RQ) based on the method described by [15, 28]. Furthermore, the fluconazole RQ was based on the environmental predicted no-effect concentration (PNEC-ENV) values reported by [29]. The RQ ≥1 denotes high risk, <1 denotes medium risk and <0.1 denotes low risk. The RQ is calculated using the following equation:

$$RQ = \frac{C}{PNEC}$$

Where: RQ = Risk quotient

 C = measured antibiotic concentration

 PNEC-ENV = environmental predicted no-effect concentration

### Statistical analyses

The R software (version 4.1.3) was used for statistical analyses. Shapiro–Wilk test was performed to determine the normality of the data. If the data failed the normality test, the Wilcoxon test was performed to determine whether there were statistically significant differences between raw and treated water of each DWPF. The student's t-test was performed if data were normally distributed. The data between raw and treated water was considered significant when the p-value was <0.05. The correlation between bacterial communities and environmental factors was determined using Redundancy analysis (RDA) using Canoco Software (Version 4.5) and pairwise correlation matrices using Statistica Software (Version 14.0.1.25). Significant differences were established at p-value <0.05. Furthermore, correlations were based on the end-point PCR analysis data [25].

## Results

### Method validation

The ARGs and 16S rRNA method was validated for amplification linearity and efficiency. The serial dilution method with known number of copies was used to construct standard curves. The standard curves had determination efficiency ($R^2$) >0.97 and efficiencies ranging between

**Table 1. Method validation parameters for the quantification of antimicrobial residues.**

| Target residues | LOD (µg/L) | LOQ (µg/L) | $R^2$ | Intra-day precision (%RSD) | | | Inter-day precision (%RSD) | Recovery (%) |
|---|---|---|---|---|---|---|---|---|
| | | | | Day 1 | Day 2 | Day 3 | | |
| Ampicillin | 0.24 | 0.79 | 0.999 | 5 | 4 | 5 | 5 | 90 |
| Trimethoprim | 0.04 | 0.13 | 0.994 | 3 | 1 | 3 | 11 | 101 |
| Ciprofloxacin | 0.11 | 0.36 | 0.998 | 4 | 2 | 3 | 13 | 102 |
| Sulfamethoxazole | 0.34 | 1.14 | 0.997 | 6 | 9 | 2 | 20 | 93 |
| Fluconazole | 0.30 | 1.01 | 0.999 | 5 | 2 | 1 | 19 | 98 |

$R^2$ = R square; LOD = limit of detection; LOQ = Limit of quantification; RSD = relative standard deviation.

90% to 110%. These parameters were regarded as reliable for determining unknown concentrations of genes in water samples [25].

Table 1 shows the parameter for method validation of antimicrobial residues. The linearity was above 0.99, LOD and LOQ ranges were 0.04–0.34 µg/L and 0.13–1.14 µg/L, respectively. The relative standard deviation (RDS) of inter-day and intraday precisions ranged between 1.09 and 19.92 µg/L. The accuracy determined by the recovery was between 90% and 102%.

## Physicochemical parameters analysis

Tables 2 to 4 show the physicochemical parameters measured in NW-E and NW-C DWPFs. In the current study, higher TDS, salinity, phosphate, nitrate, nitrite and COD levels were anticipated in raw than in treated water of NW-E and NW-C DWPFs. In NW-E DWPF, this was the case for levels of TDS (raw water = 549.69 mg/L and treated water = 548.50 mg/L), salinity (raw water = 342.06 mg/L and treated water = 340.25 mg/L), phosphate (raw water = 6.48 mg/L and treated water = 1.73 mg/L), nitrate (raw water = 0.85 mg/L and treated water = 0.80 mg/L) and nitrite (raw water = 2.44 mg/L and treated water = 1.67 mg/L), as shown in Tables 2 and 4. Significant differences (p-value <0.05) existed between raw and treated water for phosphate and nitrite levels. However, it was not the case for the COD, which was slightly higher in treated (13.00 mg/L) than in raw water (11.33 mg/L). Significant differences in COD levels were not determined due to a lack of replicates.

In NW-C DWPF, the anticipated results were true for nitrite (raw water = 2.33 mg/L and treated water = 1.67 mg/L) and COD (raw water = 31.00 mg/L and treated water = 5.00 mg/L), as shown in Table 4. No significant differences existed for the nitrite levels. The anticipation was not true for the levels of the TDS (raw water = 406.25 mg/L and treated water = 409.00 mg/L), salinity (raw water = 250.38 mg/L and treated water = 252.75 mg/L), phosphate (raw water = 3.47 mg/L and treated water = 4.21 mg/L) and nitrate (raw water = 1.46 mg/L and treated water = 1.98 mg/L). Only nitrate levels were significantly (p-value <0.05) higher in treated compared to raw water.

Temperature fluctuated between raw and treated water in NW-E and NW-C. In NW-E, the temperature range was between 10.3˚C and 24.1˚C. In NW-C, the range was between 13.6˚C and 20.9˚C. (Tables 2 and 3). The pH was higher in raw than in treated water in both NW-E and NW-C. In NW-E, pH ranged from 7.84 to 9.39, with significant differences (p-value <0.05) between raw and treated water (Table 2). In NW-C, water samples had a pH range of 7.86 and 8.1, with no significant differences (p-value >0.05) between raw and treated water (Table 3).

The physicochemical parameters were compared to the [23] for domestic water use. In NW-E treated water, pH, nitrate and nitrite levels were within the target water quality range of

**Table 2. Physical parameters with standard deviations measured in 2020, 2021 and 2022 in raw and treated water of NW-E DWPF.**

| Date | Sites | TDSs (mg/L) | pH | Salinity (mg/L) | Temperature (°C) |
|---|---|---|---|---|---|
| | DWAF (1996a) | ≤450 | ≥6.0 to ≤9.0 | ≤100 | - |
| Dec 2020 | Raw inlet 1 | 643.00±1.00 | 8.20±0.05 | 306.67±0.58 | 23.10±0.10 |
| | Raw inlet 2 | 670.67±0.67 | 8.35±0.02 | 322.00±0.00 | 22.83±0.06 |
| | Treated water | 651.67±2.08 | 7.90±0.02 | 311.33±1.53 | 23.40±0.66 |
| Feb 2021 | Raw inlet 1 | 619.67±1.65 | 8.44±0.15 | 302.67±0.58 | 24.10±0.20 |
| | Raw inlet 2 | 646.33±0.58 | 8.02±0.04 | 315.33±0.58 | 23.60±0.00 |
| | Treated water | 633.33±1.53 | 7.87±0.00 | 304.33±0.58 | 24.37±0.38 |
| Jul 2021 | Raw inlet 1 | 524.67±1.53 | 9.38±0.02 | 353.00±1.00 | 13.27±0.21 |
| | Raw inlet 2 | 543.00±2.00 | 9.39±0.06 | 360.67±0.58 | 10.33±0.32 |
| | Treated water | 532.00±1.00 | 8.96±0.01 | 357.67±0.58 | 12.13±0.47 |
| Aug 2021 | Raw inlet 1 | 527.00±1.00 | 8.55±0.01 | 356.33±0.58 | 14.27±0.25 |
| | Raw inlet 2 | 549.00±1.00 | 8.30±0.03 | 370.67±0.58 | 14.10±0.46 |
| | Treated water | 528.00±1.00 | 8.41±0.02 | 356.00±1.00 | 13.87±0.35 |
| Sep 2021 | Raw inlet 1 | 529.00±2.65 | 8.58±0.03 | 361.33±0.58 | 17.80±0.20 |
| | Raw inlet 2 | 543.67±1.53 | 8.40±0.01 | 371.33±0.58 | 17.67±0.42 |
| | Treated water | 533±0.67 | 8.36±0.02 | 369.67±3.79 | 17.87±0.15 |
| Dec 2021 | Raw inlet 1 | 484.33±0.58 | 8.41±0.03 | 331.67±1.53 | 22.90±0.10 |
| | Raw inlet 2 | 488.67±1.15 | 7.84±0.02 | 332.67±1.15 | 20.77±0.67 |
| | Treated water | 486.67±1.15 | 8.09±0.02 | 333.67±1.53 | 22.10±0.26 |
| Feb 2022 | Raw inlet 1 | 499.33±0.58 | 8.09±0.04 | 342.33±0.58 | 21.57±0.06 |
| | Raw inlet 2 | 510.00±1.00 | 7.98±0.02 | 348.67±2.52 | 22.43±0.15 |
| | Treated water | 505.33±0.33 | 7.84±0.01 | 344.67±0.58 | 22.33±1.53 |
| Mar 2022 | Raw inlet 1 | 512.67±0.58 | 8.34±0.01 | 350.00±0.00 | 19.40±1.15 |
| | Raw inlet 2 | 506.67±0.58 | 8.23±0.01 | 346.67±0.58 | 20.30±0.20 |
| | Treated water | 517.67±0.58 | 8.17±0.01 | 353.67±0.58 | 19.90±0.10 |

TDSs- total dissolved solids; "-"–not specified

[23]. However, salinity and TDS levels were not. These levels were 240.25 mg/L (salinity) and 98,50 mg/L (TDS) higher than the [23] values. In NW-C treated water, pH, TDS, nitrate and nitrite levels were within the target water quality range of [23]. However, salinity levels were not. These levels were 150 mg/L higher than the [23] values.

## Bacterial community composition

Fig 1 represents phylum-level changes in the bacterial community composition of NW-E and NW-C DWPFs. NW-E and NW-C DWPFs had seven phyla with >2% relative abundance. The following trends were observed: In NW-E DWPF, the following phyla dominated: Proteobacteria (raw water = 38.22% and treated water = 40.22%), Bacteroidota (raw water = 20.05% and treated water = 14.71%), Acidobacteriota (raw water = 18.33% and treated water = 10.71%) and Cyanobacteria (raw water = 9.98% and treated water = 10.32%). In NW-C DWPF, the following phyla dominated: Proteobacteria (raw water = 48.50% and treated water = 50.64%), Bacteroidota (raw water = 21.99% and treated water = 14.21%) and Acidobacteriota (raw water = 7.67% and treated water = 7.49%).

In NW-E and NW-C DWPFs, there were 12 genera with >1% relative abundance, as shown in Fig 2. The following trends were observed: In NW-E DWPF, the following genera dominated: *Comamonas* (raw water = 11.84% and treated water = 5.13%), *Dickeya* (raw

**Table 3. Physical parameters with standard deviations measured in 2020, 2021 and 2022 in raw and treated water of NW-C DWPF.**

| Date | Sites | TDSs (mg/L) | pH | Salinity (mg/L) | Temperature (˚C) |
|---|---|---|---|---|---|
| | DWAF (1996a) | ≤450 | ≥6.0 to ≤9.0 | ≤100 | - |
| Dec 2020 | Raw inlet | 478.67±1.15 | 8.10±0.04 | 232.00±0.00 | 20.47±0.35 |
| | Treated water | 476.33±2.08 | 7.94±0.01 | 231.33±0.58 | 20.87±0.21 |
| Feb 2021 | Raw inlet | 483.33±0.58 | 8.15±0.05 | 233.33±0.58 | 20.07±0.06 |
| | Treated water | 490.67±0.58 | 7.89±0.01 | 236.33±0.58 | 18.77±0.21 |
| Jul 2021 | Raw inlet | 378.00±2.00 | 9.31±0.03 | 249.33±1.15 | 13.63±0.31 |
| | Treated water | 381.67±3.21 | 9.28±0.06 | 254.00±1.00 | 13.80±0.40 |
| Aug 2021 | Raw inlet | 381.67±0.58 | 8.29±0.01 | 256.00±1.00 | 15.93±0.06 |
| | Treated water | 382.67±0.58 | 8.22±0.03 | 257.67±0.58 | 17.33±0.06 |
| Sep 2021 | Raw inlet | 378.33±1.15 | 8.36±0.03 | 254.00±1.00 | 15.83±0.25 |
| | Treated water | 371.67±9.29 | 8.35±0.03 | 255.33±1.15 | 15.07±0.15 |
| Dec 2021 | Raw inlet | 381.33±0.58 | 8.20±0.01 | 259.67±0.58 | 19.87±0.15 |
| | Treated water | 389.00±1.00 | 8.25±0.02 | 263.33±1.15 | 19.63±0.31 |
| Feb 2022 | Raw inlet | 386.67±0.58 | 8.15±0.02 | 262.00±1.00 | 17.80±0.20 |
| | Treated water | 391.67±0.58 | 8.12±0.02 | 264.00±0.00 | 18.23±0.31 |
| Mar 2022 | Raw inlet | 382.00±0.00 | 8.34±0.01 | 257.00±0.00 | 15.67±0.06 |
| | Treated water | 386.67±0.58 | 8.16±0.01 | 260.67±0.58 | 16.47±0.58 |

TDSs- total dissolved solids; "-"–not specified

water = 6.67% and treated water = 8.47%), *Nocardia* (raw water = 3.72% and treated water = 5.44%) and *Clade III* (raw water = 5.20% and treated water = 1.42%). In NW-C DWPF, the following genera dominated: *Comamonas* (raw water = 16.28% and treated water = 5.57%), *Dickeya* (raw water = 9.91% and treated water = 10.09%), *Allorhizobium-*

**Table 4. Chemical parameters with standard deviations measured in 2021 and 2022 in raw and treated water of NW-E and NW-C DWPFs.**

| Date | Sites | | Phosphate (mg/L) | Nitrates (mg/L) | Nitrites (mg/L) | COD (mg/L) |
|---|---|---|---|---|---|---|
| | | DWAF (1996a) | - | ≤6 | ≤6 | - |
| Feb 2021 | NW-E | Raw inlet 1 | 2.69±0.53 | 0.87±0.15 | 3.00±0.00 | 0.00 |
| | | Raw inlet 2 | 2.37±0.34 | 0.77±0.42 | 2.33±1.15 | 18.00 |
| | | Treated water | 2.67±0.68 | 1.00±0.17 | 2.00±0.00 | 16.00 |
| | NW-C | Raw inlet | 1.25±0.22 | 1.67±0.12 | 1.00±0.00 | 85.00 |
| | | Treated water | 3.37±0.84 | 1.83±0.32 | 2.00±1.00 | 4.00 |
| Sep 2021 | NW-E | Raw inlet 1 | 3.51±0.84 | 0.73±0.06 | 2.00±0.00 | 13.00 |
| | | Raw inlet 2 | 4.48±2.09 | 0.60±0.00 | 3.00±0.00 | 8.00 |
| | | Treated water | 2.40±0.23 | 0.57±0.12 | 1.00±0.00 | 14.00 |
| | NW-C | Raw inlet | 3.93±0.58 | 1.77±0.06 | 2.00±0.00 | 4.00 |
| | | Treated water | 3.43±1.32 | 1.80±0.00 | 1.00±0.00 | 6.00 |
| Mar 2022 | NW-E | Raw inlet 1 | 20.08±0.12 | 1.03±0.15 | 1.33±1.15 | 13.00 |
| | | Raw inlet 2 | 5.74±2.25 | 1.07±0.06 | 3.00±0.00 | 16.00 |
| | | Treated water | 1.06±0.84 | 0.83±0.32 | 2.00±0.00 | 9.00 |
| | NW-C | Raw inlet | 5.29±2.32 | 0.93±0.15 | 4.33±1.15 | 4.00 |
| | | Treated water | 5.82±3.100 | 2.30±0.87 | 2.00±1.00 | 5.00 |

COD–Chemical oxygen demand; "-"–not specified

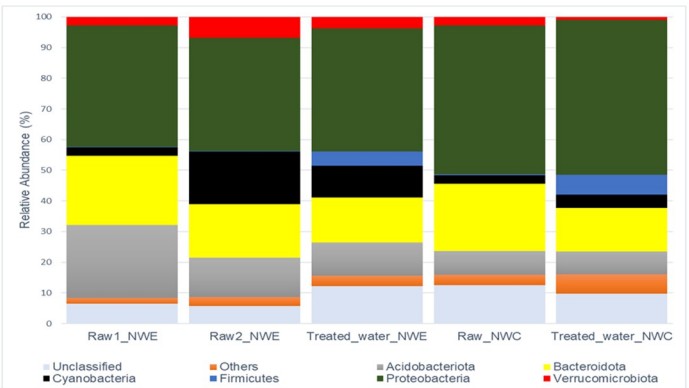

**Fig 1. Stacked column illustrating phylum-level changes in bacterial community composition of raw and treated water at NW-E and NW-C DWPFs.** Others consist of phyla with relative abundance < 2.00%.

*Neorhizobium-Pararhizobium-Rhizobium* (raw water = 9.81% and treated water = 2.03%) and Other (raw water = 3.11% and treated water = 5.73%).

The Shannon Index (species diversity and evenness), Chao1 estimator (richness), ACE estimator (richness) and Simpson index (species dominance) were calculated to determine Alpha diversity and significant differences for raw and treated water of NW-E and NW-C DWPFs as shown in Fig 3. Similar observations were made in NW-E and NW-C DWPFs in which treated water was more even, diverse and rich compared to raw water. However, more dominance was recorded for treated water in NW-C and raw water 1 for NW-E. Furthermore, no significant differences were observed (p-value >0.05) between raw and treated water.

## Screening on antibiotic resistance genes

The extracted eDNA was subjected to end-point PCR for ARGs, as shown in Fig 4. Sampling sites and ARG levels are represented by circularly arranged segments whose length is proportional to the total percentage values of ARGs. The outer rings with percentages are stacked bar plots representing a cell's relative contribution to ARG levels. Ribbons between ARG and sampling sites represent the detection percentages. This section focused on the ribbons

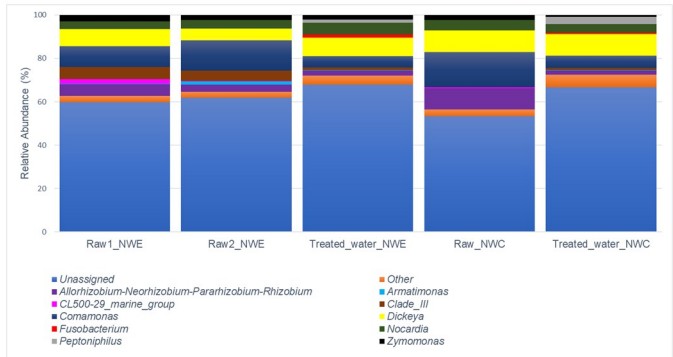

**Fig 2. Stacked column illustrating genus-level changes in bacterial community composition in raw and treated water of NW-E and NW-C DWPFs.** Others consist of genera with relative abundance < 1.00%.

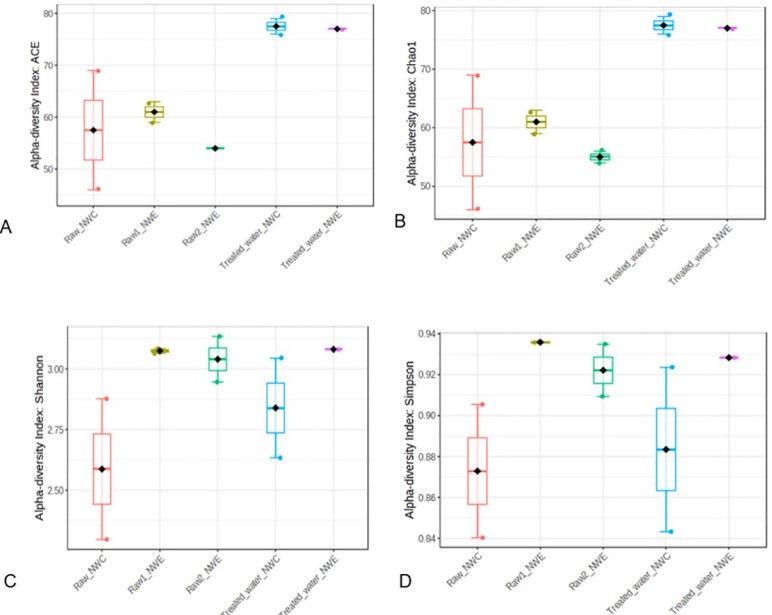

**Fig 3. Bacterial alpha diversity indices (A-ACE, B-Chao1, C-Shannon and D-Simpson) in raw and treated water of NW-E and NW-C DWPF.**

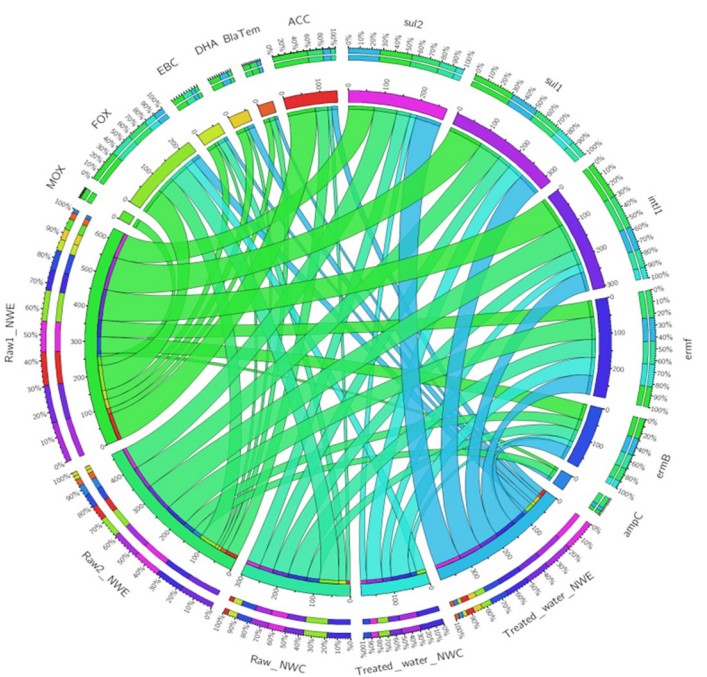

**Fig 4. Chord diagram illustrating the end-point PCR screening process of ARGs in raw and treated water of NW-E and NW-C DWPFs.**

representing the length of the total percentage values of ARGs in different sampling sites. In NW-E DWPF, the number of samples that were positive for *ampC*, *bla*$_{TEM}$, *intl1* and *sul1* genes was higher in raw than in treated water samples. However, more *sul2* positive samples were in treated water (75.00%) compared to raw water (70.83%). Furthermore, a similar number of positive samples for *ermB* (41.67%) and *ermF* (58.33%) genes was observed in raw and treated water.

In NW-C DWPF samples, *intl1*, *ermB* and *sul2* genes were detected in larger numbers in raw water compared to treated water samples (Fig 4). However, *sul1* gene was detected in higher numbers of samples of treated water (45.45%) compared to raw water (40.00%). Furthermore, the *ermF* gene was detected in similar numbers in raw and treated water samples (54.44%). The *ampC* and *bla*$_{TEM}$ genes were not in raw and treated water samples of NW-C DWPF.

A general trend was observed in which pAmpCs such as ACC, MOX, FOX and EBC were detected in more raw samples compared to treated water samples in NW-E and NW-C DWPF (Fig 4). The DHA gene detection was detected in more treated water samples of NW-E DWPF than NW-C DWPF. Another trend observed was the absence of the CIT gene in raw and treated water samples of NW-E and NW-C DWPFs.

## Quantification of antibiotic resistance genes

Table 5 represents the concentrations and standard deviation of pAmpCs, *intl1* and *sul1* genes in DWPFs. This was determined according to [25] for pAmpCs, [24]. However, samples in which these ARGs were not detected during the end-point PCR screening process were excluded. The *sul1* concentrations were higher in treated water of NW-E and NW-C DWPFs. In NW-E, the *sul1* concentrations were $3.73 \times 10^{-4}$ copies/16S rRNA in treated water and $2.53 \times 10^{-4}$ copies/16S rRNA in raw water. In NW-C, the *sul1* concentrations were $6.03 \times 10^{-2}$ copies/16S rRNA in raw water and $7.97 \times 10^{-2}$ in treated water.

In NW-E DWPF, the EBC concentrations were higher in treated water ($6.37 \times 10^{-2}$ copies/16S rRNA) than in raw water ($6.37 \times 10^{-2}$ copies/16S rRNA), as shown in Table 5. In NW-C DWPF, the EBC concentrations were $7.67 \times 10^{-2}$ copies/16S rRNA in raw water. However, treated water was not subjected to real-time PCR analysis because it was not detected during the screening using end-point PCR.

In NW-E DWPF, the ACC concentrations in treated water were $7.32 \times 10^{-2}$ copies/16S rRNA and $1.58 \times 10^{-1}$ copies/16S rRNA in raw water (Table 5). This was inconsistent with the

**Table 5. Concentrations of ARGs in raw and treated water of NW-E and NW-C DWPFs (units in gene copies/16S Rrna ± standard deviation).**

| | Sites | EBC | ACC | CIT | DHA | MOX | FOX | *IntI1* | *Sul1* |
|---|---|---|---|---|---|---|---|---|---|
| NW-E | Raw water 1 | $2.15 \times 10^{-2}$ $\pm 2.68 \times 10^{-2}$ | $2.68 \times 10^{-1}$ $\pm 2.31 \times 10^{-1}$ | - | $2.26 \times 10^{-2}$ $\pm 3.62 \times 10^{-2}$ | $1.29 \times 10^{-1}$ $\pm 2.03 \times 10^{-1}$ | $1.04 \times 10^{0}$ $\pm 7.65 \times 10^{-1}$ | $2.03 \times 10^{-4}$ $\pm 7.22 \times 10^{-5}$ | $1.95 \times 10^{-3}$ $\pm 1.43 \times 10^{-3}$ |
| | Raw water 2 | $5.23 \times 10^{-2}$ $\pm 1.46 \times 10^{-2}$ | $4.79 \times 10^{-2}$ $\pm 1.38 \times 10^{-2}$ | - | $4.18 \times 10^{-3}$ $\pm 1.81 \times 10^{-3}$ | - | $2.01 \times 10^{-1}$ $\pm 1.28 \times 10^{-1}$ | $5.76 \times 10^{-4}$ $\pm 9.99 \times 10^{-4}$ | $3.10 \times 10^{-3}$ $\pm 3.63 \times 10^{-3}$ |
| | Treated water | $6.37 \times 10^{-2}$ $\pm 2.27 \times 10^{-2}$ | $7.32 \times 10^{-2}$ $\pm 1.57 \times 10^{-1}$ | - | $4.16 \times 10^{-2}$ $\pm 1.70 \times 10^{-2}$ | - | $4.12 \times 10^{-1}$ $\pm 6.49 \times 10^{-1}$ | $6.12 \times 10^{-4}$ $\pm 3.76 \times 10^{-4}$ | $3.73 \times 10^{-3}$ $\pm 1.04 \times 10^{-3}$ |
| NW-C | Raw water | $7.67 \times 10^{-2}$ $\pm 2.33 \times 10^{-2}$ | $3.85 \times 10^{-2}$ $\pm 1.60 \times 10^{-2}$ | - | - | - | $1.44 \times 10^{1}$ $\pm 1.33 \times 10^{1}$ | $3.50 \times 10^{-4}$ $\pm 1.89 \times 10^{-6}$ | $6.03 \times 10^{-2}$ $\pm 2.02 \times 10^{-2}$ |
| | Treated water | - | <LOD | - | - | - | $6.94 \times 10^{-2}$ $\pm 1.42 \times 10^{-1}$ | ND | $7.97 \times 10^{-2} \pm .81 \times 10^{-2}$ |

LOD–limit of detection; ND–not done; "-"–not detected by end-point PCR

finding obtained in NW-C DWPF, where the ACC concentrations were higher in raw water ($3.85 \times 10^{-2}$ copies/16S rRNA) than in treated water (<LOD).

In NW-E DWPF, the DHA concentrations were higher in treated water ($4.16 \times 10^{-2}$ copies/16S rRNA) compared to raw water ($1.34 \times 10^{-2}$ copies/16S rRNA), as shown in Table 5. Since the DHA gene was not detected during the end-point PCR screening process in raw and treated water, it was not subjected to real-time PCR analysis.

In NW-E DWPF, the *intl1* concentrations were higher in treated water ($6.12 \times 10^{-4}$ copies/16S rRNA) than in raw water ($3.90 \times 10^{-4}$ copies/16S rRNA). In NW-C DWPF, the *intl1* concentrations were $3.50 \times 10^{-4}$ copies/16S rRNA in raw water. However, the *intl1* concentrations of treated water were not determined due to insufficient eDNA.

The FOX concentrations were higher in raw water of NW-E and NW-C DWPFs. In NW-E, the FOX concentrations in $4.12 \times 10^{-1}$ copies/16S rRNA in treated water and $6.22 \times 10^{-1}$ copies/16S rRNA in treated water (Table 5). In NW-C DWPF, the FOX concentrations were $6.94 \times 10^{-2}$ copies/16S rRNA in treated water and $1.44 \times 10^{1}$ copies/16S rRNA in treated water.

In NW-E, the MOX concentrations were $1.29 \times 10^{-1}$ copies/16S rRNA in raw water 1 (Table 5). However, the MOX concentrations were not determined in raw water 2 and treated water of NW-E due to its absence during the end-point PCR screening process. The MOX concentrations were also determined in NW-C raw and treated water. Furthermore, the CIT concentrations in NW-E and NW-C were not determined for the above reasons.

## Quantification of antibiotic residues

The antimicrobial concentrations (μg/L) and standard deviations are shown in Tables 6 and 7. In NW-C DWPF, ampicillin, ciprofloxacin, sulfamethoxazole and fluconazole concentrations were higher in raw than treated water. However, trimethoprim concentrations were higher in treated than in raw water. In NW-E DWPF, all target compound concentrations were higher in raw than treated water. Furthermore, in NW-E and NW-C DWPFs, no significant differences (p-value >0.05) existed between raw and treated water.

In DWPFs, ampicillin concentrations were higher in raw water (NW-E range between 4.20 and 27.77 μg/L and NW-C range between 2.48 and 16.93 μg/L) than in treated water (NW-E range between 2.83 and 14.19 μg/L and NW-C range between 2.43 and 15.72 μg/L). However, in November 2020, in NW-E, ampicillin concentrations were higher in treated water (11.53 μg/L) than in raw water (10.23 μg/L). In NW-C, higher ampicillin concentrations in treated water were also measured in August 2021 and February 2022.

In NW-C DWPF, ciprofloxacin concentrations were 0.68 μg/L in raw water and 0.53 μg/L in treated water. In February 2022, ciprofloxacin concentrations were 1.09 μg/L in raw and treated water of NW-C (Table 7). In NW-E DWPF, ciprofloxacin concentrations were 4.69 μg/L in raw water and 1.02 μg/L in treated water (Table 6).

Sulfamethoxazole concentrations were 1.22 μg/L in raw water and 0.78 μg/L in treated water in NW-E DWPF (Table 6). However, in February 2022, sulfamethoxazole concentrations were higher in treated water (2.64 μg/L) than in raw water (2.15 μg/L). In NW-C DWPF, sulfamethoxazole concentrations were 0.95 μg/L in raw water and 0.75 μg/L in treated water (Table 7).

In NW-E DWPF, trimethoprim concentrations were higher in treated water in November 2020 and August 2021 (Table 6). Overall, trimethoprim concentrations were higher in raw water (0.20 μg/L) than in treated water (0.19 μg/L). Contrary, in NW-C DWPF, trimethoprim concentrations were higher in treated water (0.10 μg/L) than in raw water (0.07 μg/L), as shown in Table 7.

**Table 6. Concentrations with standard deviations of target antimicrobial residues in raw and treated water of NW-E DWPF.**

| Date | Sampling sites | Ampicillin (µg/L) | Ciprofloxacin (µg/L) | Sulfamethoxazole (µg/L) | Trimethoprim (µg/L) | Fluconazole (µg/L) |
|---|---|---|---|---|---|---|
| Oct-20 | Raw inlet 1 | 5.95±0.84 | 3.43±0.41 | 3.46±0.52 | <LOQ | <LOQ |
| | Raw inlet 2 | 12.97±0.12 | 4.91±0.73 | 3.78±0.13 | <LOQ | <LOQ |
| | Treated water | 8.10±0.85 | 3.42±0.28 | 2.85±0.49 | <LOQ | <LOQ |
| Nov-20 | Raw inlet 1 | 8.90±1.13 | <LOQ | 1.53±0.37 | 0.18±0.00 | <LOQ |
| | Raw inlet 2 | 11.55±2.09 | <LOQ | <LOQ | <LOQ | <LOQ |
| | Treated water | 11.53±4.15 | <LOQ | <LOQ | 0.19±0 | <LOQ |
| Dec-20 | Raw inlet 1 | 16.85±3.45 | 0.95±0.18 | 2.03±0.37 | 0.82±0.04 | <LOQ |
| | Raw inlet 2 | 24.58±4.59 | 0.74±0.23 | 2.00±0.13 | 0.24±0.01 | <LOQ |
| | Treated water | 12.56±0.37 | 0.83±0.18 | <LOQ | 0.20±0.01 | <LOQ |
| Feb-21 | Raw inlet 1 | 25.81±1.81 | 2.25±0.05 | <LOQ | <LOQ | <LOQ |
| | Raw inlet 2 | 11.79±2.37 | 2.83±0.74 | <LOQ | 0.17±0.03 | <LOQ |
| | Treated water | 10.02±0.04 | 0.60±0.04 | <LOQ | <LOQ | <LOQ |
| Jul-21 | Raw inlet 1 | 4.20±0.17 | 1.32±0.20 | <LOQ | 0.40±0.03 | <LOQ |
| | Raw inlet 2 | 6.10±2.34 | 0.87±0.01 | <LOQ | 0.22±0.02 | <LOQ |
| | Treated water | 2.83±0.38 | 0.55±0.07 | <LOQ | 0.25±0.01 | <LOQ |
| Aug-21 | Raw inlet 1 | 27.77±1.29 | 1.37±0.30 | <LOQ | 0.40±0.01 | <LOQ |
| | Raw inlet 2 | 11.92±0.06 | <LOQ | <LOQ | <LOQ | <LOQ |
| | Treated water | 3.27±0.10 | 1.00±0.06 | <LOQ | 0.39±0.05 | <LOQ |
| Feb-22 | Raw inlet 1 | 18.89±3.81 | 46.52±1.47 | 2.64±0.26 | 0.35±0.02 | 9.49±0.78 |
| | Raw inlet 2 | 20.08±3.68 | 0.44±0.00 | 1.65±0.12 | 0.33±0.01 | 19.03±5.78 |
| | Treated water | 14.18±3.81 | 0.75±0.09 | 2.64±0.12 | 0.32±0.04 | 8.34±1.50 |

LOQ–limit of quantification

In NW-E and NW-C DWPFs, fluconazole concentrations were <LOQ from October 2020 until August 2021 (Tables 6 and 7). In February 2022, fluconazole concentrations were 14.26 µg/L in raw water and 8.34 µg/L in treated water in NW-E DWPF. Furthermore, in NW-C DWPF, fluconazole concentrations were 27.71 µg/L in raw water and 13.84 µg/L in treated water.

**Table 7. Concentrations with standard deviations of target antimicrobial residues in raw and treated water of NW-C DWPF.**

| Date | Sites | Ampicillin (µg/L) | Ciprofloxacin (µg/L) | Sulfamethoxazole (µg/L) | Trimethoprim (µg/L) | Fluconazole (µg/L) |
|---|---|---|---|---|---|---|
| Nov-20 | Raw inlet | 16.93±3.69 | <LOQ | <LOQ | 0.16±0.01 | <LOQ |
| | Treated water | 15.72±0.08 | <LOQ | <LOQ | 0.16±0.03 | <LOQ |
| Dec-20 | Raw inlet | 5.04±0.41 | <LOQ | 3.51±0.14 | <LOQ | <LOQ |
| | Treated water | 3.91±1.34 | <LOQ | 2.64±0.95 | 0.25±0.05 | <LOQ |
| Feb-21 | Raw inlet | 10.50±2.06 | 2.42±0.27 | <LOQ | <LOQ | <LOQ |
| | Treated water | 7.08±1.29 | 2.08±0.28 | <LOQ | <LOQ | <LOQ |
| Jul-21 | Raw inlet | 2.48±0.95 | 0.56±0.07 | <LOQ | 0.24±0.03 | <LOQ |
| | Treated water | 2.43±0.76 | <LOQ | <LOQ | 0.21±0.02 | <LOQ |
| Aug-21 | Raw inlet | 2.75±0.09 | <LOQ | <LOQ | <LOQ | <LOQ |
| | Treated water | 3.15±0.09 | <LOQ | <LOQ | <LOQ | <LOQ |
| Feb-22 | Raw inlet | 6.65±0.65 | 1.09±0.06 | 2.17±0.05 | <LOQ | 27.71±9.06 |
| | Treated water | 7.45±0.15 | 1.09±0.06 | 1.91±0.31 | <LOQ | 13.84±1.12 |

LOQ–limit of quantification

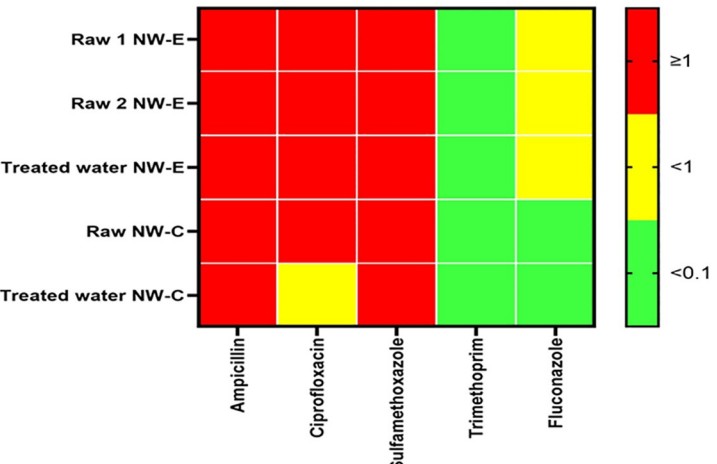

**Fig 5. Heatmap of risk quotients (RQs) in raw and treated water of NW-E and NW-C DWPFs.**

## Risk quotient

The risk quotients of raw and treated water in NW-E and NW-C DWPFs are shown in Fig 5. A general trend observed in NW-E and NW-C DWPFs was that the RQ of sulfamethoxazole, ampicillin and ciprofloxacin was ≥1 in both raw and treated water. However, the RQ of ciprofloxacin in NW-C treated water which was <1. In NW-E, fluconazole was <1 in raw and treated water; in NW-C, it was <0.1 in both water types. Another trend observed was the RQ of trimethoprim was <0.1 in NW-E and NW-C.

## Correlation of bacterial communities with ARGs, antibiotics and physicochemical parameters in drinking water production facilities

The RDA correlation of bacterial communities with physicochemical parameters, ARGs and antimicrobial residues is shown in Fig 6. Initially, commonly prescribed antimicrobial agents

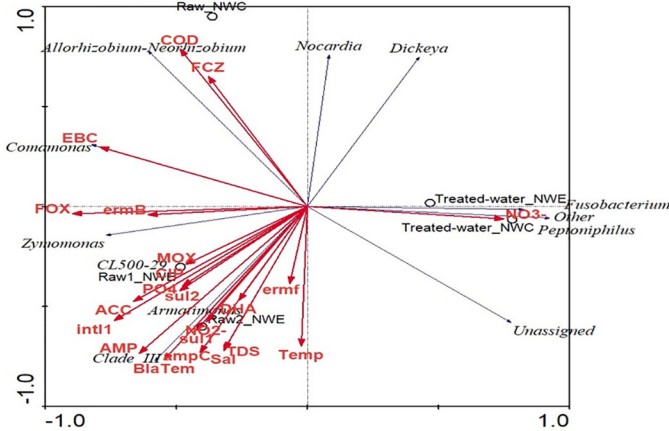

**Fig 6. RDA of correlation between (A) bacterial community composition, physicochemical parameters, ARGs and antibiotic residues in raw and treated water of NW-E and NW-C DWPFs.** FCZ–fluconazole; temp–temperature; AMP–ampicillin; sal–salinity; COD–Chemical oxygen demand, TDS–Total Dissolved Solids; SMX–Sulfamethoxazole; NO2⁻—nitrite, NO3⁻—nitrate; PO4 –phosphate; TMP–trimethoprim.

and prevalent ARGs were included in the analysis [6]. However, due to the failed optimization for screening ARGs and quantifying antimicrobial residues, only a few parameters were included in the current study. The results revealed no significant correlation (p-value >0.05) in the first and other canonical axes. Furthermore, the pairwise correlation between bacterial communities and environmental factors is shown in S1 Appendix. *Dickeya* and *Armatimonas* were significantly correlated (p-value <0.05) with the *ampC* gene. *CL500-29 marine group* had a significant correlation (p-value <0.05) with phosphate, pH, MOX gene, ACC gene, sulfamethoxazole and ciprofloxacin levels. *Clade III* significantly correlated with the $Bla_{TEM}$ gene, ACC gene and ampicillin. *Zymomonas* significantly correlated (p-value <0.05) with *ermB* and *intI1* genes.

## Discussion

In NW-C DWPFs, the phosphate and nitrate levels were higher in treated than in raw water. This is attributable to the operations and configuration of NW-C DWPF, specifically, the sump used to collect water after filtration. The water is not suspended long enough to settle suspended solids. This was not the case in the DWPF situated NW-E. Furthermore, in Bayas, Ecuador [30] and Kafr El-Shinawy, Egypt [31], phosphate and nitrate levels are always lower in treated water. The latter studies focussed on the water quality of raw and treated water.

The nitrite levels were higher in raw compared to treated water of NW-E (p-value <0.05) and NW-C (p-value >0.05) DWPFs. The findings of NW-E DWPFs could be attributed to the sedimentation process that allows suspended particles to settle out [32]. Similar nitrite trends in which nitrite levels decreased from raw to treated water were also observed in Jiangsu Province, China and Kafr El-Shinawy, Damietta, Egypt [31]. Overall, the nitrate and nitrite levels measured in NW-E and NW-C DWPFs were within the target water quality range (0–6 mg/L) of [23] for domestic use. However, the phosphate, nitrate and nitrite may promote bacterial growth and microbial growth from treated water to distribution networks [10].

Consistent with [12], TDS levels were higher in treated water than in raw water in NW-C DWPF. These levels indicate that water might be objectionable due to its flat, insipid taste [23]. Consistent with [33], in NW-E and NW-C DWPF, salinity levels were higher in raw than treated water. However, higher TDS and salinity levels in NW-E were attributed to the area's geology. This DWPF is in the dolomitic region [34]. Generally, consumption of NW-E and NW-C treated water may have a noticeable salty taste which may be displeasing for people on a sodium-redistricted diet [23].

In NW-E DWPF, the COD levels were higher in treated water than in raw water. The opposite was true for the NW-C DWPF. COD levels indicate the dissolved oxygen needed to oxidize organic matter [35]. Temperature variation in NW-E and NW-C may also contribute to the growth of heterotrophic bacterial populations [10, 36]. Temperatures were typically between 15 and 25˚C. Furthermore, the pH of treated water in NW-E and NW-C was considered alkaline, which is not uncommon in the North West Province of South Africa [12]. Overall, the pH recorded in the current study was within the target water quality range (6.0–9.0) of [23] for domestic use. It was important to achieve 8.00 to ensure optimum pH [35].

Treated water of NW-E and NW-C has no adverse human health effects and, thus, is safe for human consumption [23]. However, raw and treated water's temperature, nutrients and pH may favour microbial survival [10, 12].

Consistent with other studies, Proteobacteria was the most dominant phylum in NW-E and NW-C [18, 19, 37–39]. *Comamonas* and *Dickeya* genera were the dominant genera of Proteobacteria. Generally, *Comamonas* are not human pathogens [40]. However, consumption of

treated water in NW-E and NW-C may put immunocompromised at risk since they are likely to contract bacteremia caused by some *Comamonas* species [41, 42].

Irrigation of crops with NW-E and NW-C raw water may transmit *Dickeya* to plants since these are plant pathogens, thus leading to an economic loss of important crop plants. Furthermore, *Dickeya* species are part of the top 10 phytopathogen bacteria [43, 44].

Genera of phylum Bacteroidota were assigned as others or unclassified despite being the second abundant phylum. Genera belonging to Bacteroidota, such as *Bacteroides*, are considered human pathogens with an estimated mortality rate of 19% [45].

In NW-E and NW-C DWPF, the relative abundance of Firmicutes phylum was higher in treated than in raw water. This could be attributed to endospore formation that enables species belonging to Firmicutes to resist unfavourable conditions such as desiccation and environmental stress, specifically chlorination in NW-E and NW-C DWPFs [46]. Firmicutes have also been reported in treated water and distribution networks of Northern China DWPF [38]. In the current study, the relative abundance of the *Peptoniphilus* genus belonging to Firmicutes was higher in treated water of DWPFs. *Peptoniphilus* species may harbour ARGs and are considered opportunistic pathogens to humans [47]. Thus, consuming *Peptoniphilus* through treated water may have negative human health impacts.

In NW-E DWPF, the relative abundance of *Nocardia* of phylum Actinomycetota was higher in treated water (5.44%) than in raw water (3.72%). The findings obtained in NW-E could be attributed to *Nocardia's* ability to survive dormantly in unfavourable conditions such as exposure to chlorine [46]. *Nocardia* species are saprophytes and can also act as opportunistic pathogens [48].

In NW-E and NW-C, a trend was observed in which treated water was more species diverse, even and richer than raw water. Furthermore, species dominance was higher in treated water of NW-C. However, in NW-E, species dominance was the same in raw and treated water. The higher alpha diversity in treated water may be due to a higher relative abundance of bacterial communities than in raw water, which is associated with biofilm formation and bacterial regrowth in treated water [37] This was inconsistent with other studies that reported higher alpha diversity in raw than in treated water in DWPFs in China and Finland [39, 49]. Achieving low alpha diversity in treated water suggests that water is biologically stable [39], which was not the case for NW-E and NW-C.

The current study revealed the presence of core bacterial communities shared between NW-E and NW-C raw and treated water. The relative abundance of bacterial communities fluctuated between raw and treated water. Furthermore, it is important to emphasize that biofilm formation enables some bacteria to overcome unfavourable conditions posed by chlorine; hence higher *Nocardia* and *Peptoniphilus* were reported in treated water [46]. This could have resulted in higher alpha diversity in treated water of NW-E and NW-C observed in the current study. Furthermore, some bacterial communities are potential human and plant pathogens that could harbour ARGs. These could be transferred to the human populations if present in drinking water.

In NW-C DWPF, *ampC* and *bla*$_{TEM}$ genes were not detected. However, in NW-E DWPF, the *ampC* gene was detected in 18.18% of raw water samples but only in 10% of the treated samples. This was also seen in distribution networks in Manitoba, Canada [50].

The pAmpCs, *bla*$_{TEM}$ and *ampC* confer resistance to antibiotics belonging to the β-lactam spectrum [51]. In South Africa, β-lactam spectrum antibiotics, such as cephalosporins (15.73%) and penicillin (40.66%), constitute the highest prescribed antibiotics in private primary health care. Furthermore, South Africa tops the list of penicillin use compared to other BRICS nations, the United Kingdom and the United States of America [6]. Golsha *et al.* state that the presence of more than one β-lactam spectrum resistance gene, as shown in the current

study, may increase antibiotic resistance since they are subjected to HGT among species in a wide range of environments [51].

FOX, EBC and ACC genes were detected raw water in NW-E and NW-C DWPFs. The MOX gene was only detected in raw water 1 in NW-E. DHA was also absent in NW-C DWPF, but it was detected in raw than treated water of NW-E DWPF. Although not extensively featured in environmental studies, these genes have been reported in various niches [13, 52–54]. pAmpCs are widespread in various niches where their effects may be undesirable for human health [51] and they are underexplored in the DWPFs. Furthermore, according to the searched databases, there is a lack of information about the concentrations of pAmpCs in DWPFs that could be useful in developing quantitative risk assessments for humans and aquatic life exposure to these ARGs.

The detection of the *ermF* gene in NW-E (58.33%) and NW-C (54.55%) DWPFs was the same for raw and treated water. [2] reported *ermF* gene was detected in a larger number of treated water of DWPF in Jiaxing City, China, compared to the raw water. However, the *ermB* gene was detected in more treated water samples compared to raw water. [2] reported higher *ermB* gene detection in raw water. The *ermB* and *ermF* genes induce resistance to macrolides-lincosamides-streptogramin B antibiotics [55, 56]. The prescription of macrolides and lincosamides is 16.81% and 1.57% in South African primary healthcare [57]. Finding these genes in the drinking water ecosystems could be the result of this generous prescription of these antibiotics.

In NW-E and NW-C DWPF, *sul2* gene was detected more in raw than treated water samples. Consistent with [58], *sul1* gene was detected in more water samples than the *sul2* gene. *Sul1* and *sul2* genes induce sulphonamide resistance and may pose health threats to HIV/AIDS patients who rely on sulfamethoxazole-trimethoprim to treat *Pneumocystis* pneumonia [2, 59, 60]. Furthermore, the *sul1* gene in water serves as a gene marker for the presence of the *intI1* gene [59]. Consistent with [61], the *intI1* gene was detected in more raw water samples than treated water in NW-E and NW-C DWPF. The *intI1* gene is associated with HGT and induction of multiple drug resistance [11, 62]. Finding these genes in the environment also indicates pollution from anthropogenic sources. Thus, the current study's findings fill this literature gap in DWPFs by determining the distribution of AmpCs in raw and treated water. The *int1* and *sul1* concentrations were also determined in the current study.

The current study revealed that EBC, FOX, ACC and DHA concentrations were higher in treated than in raw water in NW-E DWPF. These were inconsistent with FOX and ACC concentrations recorded in NW-C that were higher in raw than treated water. Other pAmpCs were either absent in raw and/or treated water during the end-point PCR screening process and, as a result, excluded from the real-time PCR analysis. Thus, their distribution was not determined from raw to treated water.

In NW-E DWPF, the higher *sul1* concentrations of $3.73 \times 10^{-3}$ copies/16S rRNA were recorded in treated water and $2.53 \times 10^{-3}$ copies/16S rRNA in raw water. Similar observations were made in NW-C, where in treated water ($7.97 \times 10^{-2}$ copies/16S rRNA), higher *sul1* concentrations were recorded compared to raw water ($6.03 \times 10^{-2}$ copies/16S rRNA). These concentrations were comparable to those reported in raw water in water in Kathmandu Valley, Nepal [1], where the researchers also studied the prevalence of ARGs in drinking and environmental water sources.

The *intI1* concentrations in NW-E ($3.90 \times 10^{-4}$ copies/16S rRNA) and NW-C ($3.50 \times 10^{-4}$ copies/16S rRNA) were comparable. Furthermore, the *intI1* concentrations in NW-E treated water were $6.12 \times 10^{-4}$ copies/16S rRNA. However, they were lower than those reported in the source water of Kathmandu Valley, Nepal [1]. Furthermore, the *intI1* concentrations in NW-C treated water were not determined.

The current study showed that environmental settings, in addition to clinical settings, are important for monitoring ARGs. Thus, the consumption of treated water harbouring ARGs is a concern for individuals in NW-E and NW-C [11]. It is also a concern to report higher detection rates and concentrations of ARGs in treated than in raw water. This could be associated with a lack of monitoring and regulation of ARGs in treated water and DWPFs not designed to remove ARGs [61]. These free-range ARGs in drinking water systems could have negative human health impacts. It could result in expensive antibiotic therapy and extended hospitalization to treat common infections if these are caused by resistant microorganisms [4, 5].

In NW-E and NW-C DWPFs, the ampicillin concentrations were higher in raw water than in treated. However, no studies were found in the searched databases addressing ampicillin residues in DWPFs. The ampicillin concentrations recorded in NW-E and NW-C were lower than those in surface water collected in Cluj-Napoca, Romania [27]. As mentioned above, the use of β-lactam spectrum antibiotics in South Africa is high due to their importance in treating infections in neonatal, pediatric and adult patients [57, 60]. However, the effectiveness of ampicillin may be affected due to the presence of β-lactamase genes that have also been reported in the present study.

Consistent with Al-Rasheed and Al-Wihda DWPFs in Baghdad City, Iraq [63], In NW-E and NW-C DWPFs, the ciprofloxacin was lower in treated than in raw water. Ciprofloxacin, belonging to fluoroquinolones, is commonly used for the treatment of multidrug-resistant bacteria, which render resistant to other classes of antibiotics such as β-lactam, aminoglycosides and macrolide [63]. Thus, it is of great concern to report ciprofloxacin in NW-E and NW-C, where bacteria could induce its resistance.

Sulfamethoxazole concentrations were higher in raw than in treated water. A similar trend was observed in other studies [64, 65]. The presence of sulfamethoxazole in water may be attributed to the 3.16% prescription of sulphonamides in South African private primary health care [57]. However, as shown in the current study, sulfonamide resistance genes are present in treated water. This may limit the treatment options listed by [60].

Trimethoprim concentrations were higher in treated water than in raw water in NW-C DWPF. Furthermore, other studies reported reduced trimethoprim concentrations from raw to treated water in DWPFs in China [64, 66]. However, [27] reported high trimethoprim concentrations of 400 µg/L in surface water. Trimethoprim usage ranks among the highest-used antibiotics in South Africa [6]. However, due to the lack of antibiotic usage for each province, it is difficult to explain the low trimethoprim concentrations in the current study.

Generally, fluconazole concentrations were <LOQ for most sampling months NW-E and NW-C DWPFs. However, in February 2022, fluconazole concentrations were higher in raw and treated water. The spike in fluconazole concentrations in February 2022 is unknown. Fluconazole concentrations have been reported in surface water in North West Province, South Africa [7]. The latter studies focussed on antifungal agents, yeast abundance and diversity in surface water. The spread of fluconazole in the water environment may be associated with its prescription for individuals living with HIV to treat candidaemia and cryptococcal meningitis [7, 67]. Its frequent application and recalcitrance to biodegradation in conventional wastewater treatment plants may lead to its prevalence in the water environment which may pose risks to water users [7].

The current study further revealed that antimicrobial residues might pose ecological risks in addition to human health based on the RQ. Ciprofloxacin, ampicillin and sulfamethoxazole were deemed high risk in NW-E and NW-C. However, in NW-C treated water, ciprofloxacin was deemed a medium risk. These were in accordance with results obtained in Chinese and European surface water, reclaimed water and groundwater where ciprofloxacin and

sulfamethoxazole were deemed high risk [14, 68, 69]. However, the ampicillin RQ of NW-E and NW-C were inconsistent with those reported in Hanoi lakes, Vietnam [15].

In NW-C DWPFs, trimethoprim and fluconazole were deemed low risk. However, in NW-E and DWPF, trimethoprim was deemed low risk and fluconazole high risk. The current study showed that the low to medium risk in treated water is not a result of the ability of DWPFs to treat antibiotics but the RQ recorded in raw water. Thus, it is important to include antifungals and antibiotics to monitor and regulate residues in water to protect ecological and human health.

Although most antimicrobial concentrations were lower in treated than raw water, no significant differences were observed (p-value >0.05). This may be due to the design of NW-E and NW-C not accounting for the removal of antimicrobial residues. The presence of these target compounds in subinhibitory concentrations may induce antibiotic resistance, as shown by the presence of ARGs in the current study. Since there is a high number of immunocompromised patients in South Africa, the effect of antibiotic resistance may be more severe [8]. Considering that ARGs may be spread through environmental settings, monitoring and regulating antibiotic and antifungal residues in water is important to reduce the spread of antimicrobial resistance through the food chain.

Antibiotic residues, physicochemical parameters and ARGs might prompt the selection of the appropriate bacterial community and subsequently change the microbial composition and diversity [70, 71]. No significant differences (p-value >0.05) were observed between bacterial communities and environmental factors using RDA analyses. However, this contrasted with the results obtained for pairwise correlation matrices. The kind of significant correlation (p-value <0.05) observed in the current study between bacterial communities and antibiotic residues led to the speculation that various responses of bacterial communities to antibiotic residues in the environment select for antibiotic resistance in opportunistic bacteria [70]. Consistent with other studies, the correlation between bacterial communities and ARGs observed in the current study indicates that bacteria are the major sources and spreaders of ARGs [70, 71]. The physicochemical parameters in NW-E and NW-C DWPF create favourable conditions for the survival of bacteria and may directly or indirectly influence the variation and levels of ARGs [72]. The operational activities of DWPF affect physicochemical parameters and bacterial communities, thus leading to a shift in alpha diversity from raw to treated water.

## Conclusion

The current study demonstrated the presence and levels of antibiotic resistance genes and antibiotic residues in two drinking water production facilities (NW-E and NW-C) in the North West Province, South Africa. It also shows how bacterial communities' composition is linked to ARGs, antibiotic residues and physicochemical parameters. This study also demonstrated that ARGs, antibiotic residues and fluconazole (antifungal agent) pollute and may affect water quality. Yet, ARGs and antimicrobial agents are not routinely monitored and regulated in DWPFs. This has led to unknown amounts of ARGs, and antibiotic residues being spread to consumers via the drinking water chain. Thus, inducing antibiotic resistance and posing public health to consumers, especially immunocompromised people. The effects of antibiotic resistance could have dismal effects in South Africa since there is a high number of PLWH (with 13.7% of the population infected with HIV/AIDS) and dilapidating public health care facilities [8]. Furthermore, some of these pollutants pose ecological risks. Thus, including environmental studies such as this one for policymaking in combating antibiotic resistance is of utmost importance.

## Supporting information

**S1 Table. Oligonucleotide primers for the end-point PCR amplification of *ampC, bla*$_{TEM}$, *IntI1*, *ermB*, *ermF*, *Sul1* and *Sul2* genes.** F- Forward primer and R- Reverse primer. (DOCX)

**S2 Table. Primers: Oligonucleotide primers for the end-point PCR amplification of six pAmpCs, namely ACC, EBC, DHA, FOX, CIT and MOX gene: F- Forward primer and R- Reverse primer.** (DOCX)

**S3 Table. Oligonucleotide primers for the real-time PCR quantification of 16S rRNA, *sul1*, *ermB* and *IntI1*: F- Forward primer and R- Reverse primer.** (DOCX)

**S1 Appendix.** (XLSX)

## Acknowledgments

The authors acknowledge Mabeo Refilwe, Abram Mahlatsi, Raimi Adekunle and Prudent Mokgokong.

## Author Contributions

**Conceptualization:** Karabo Tsholo, Lesego Gertrude Molale-Tom, Cornelius Carlos Bezuidenhout.

**Data curation:** Karabo Tsholo, Lesego Gertrude Molale-Tom, Suranie Horn, Cornelius Carlos Bezuidenhout.

**Formal analysis:** Karabo Tsholo, Cornelius Carlos Bezuidenhout.

**Funding acquisition:** Lesego Gertrude Molale-Tom, Cornelius Carlos Bezuidenhout.

**Investigation:** Karabo Tsholo, Suranie Horn.

**Methodology:** Karabo Tsholo, Lesego Gertrude Molale-Tom, Suranie Horn, Cornelius Carlos Bezuidenhout.

**Project administration:** Karabo Tsholo, Lesego Gertrude Molale-Tom, Cornelius Carlos Bezuidenhout.

**Resources:** Karabo Tsholo, Lesego Gertrude Molale-Tom, Suranie Horn, Cornelius Carlos Bezuidenhout.

**Software:** Karabo Tsholo.

**Supervision:** Lesego Gertrude Molale-Tom, Suranie Horn, Cornelius Carlos Bezuidenhout.

**Validation:** Karabo Tsholo, Suranie Horn.

**Visualization:** Karabo Tsholo, Lesego Gertrude Molale-Tom, Suranie Horn, Cornelius Carlos Bezuidenhout.

**Writing – original draft:** Karabo Tsholo.

**Writing – review & editing:** Karabo Tsholo, Lesego Gertrude Molale-Tom, Suranie Horn, Cornelius Carlos Bezuidenhout.

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
