## [Decision Letter · Decision Letter 0]

1 Dec 2023

PONE-D-23-28722Distribution of antibiotic resistance genes and antibiotic residues in drinking water production facilities: links to bacterial communityPLOS ONE

Dear Dr. Tsholo,

Thank you for submitting your manuscript to PLOS ONE. After careful consideration, we feel that it has merit but does not fully meet PLOS ONE’s publication criteria as it currently stands. Therefore, we invite you to submit a revised version of the manuscript that addresses the points raised during the review process.

We look forward to receiving your revised manuscript.

Kind regards,

José António Baptista Machado Soares, PhD

Academic Editor

PLOS ONE

Journal Requirements:

This work is based on the research supported in part by the National Research Foundation of South Africa Grant No. C2019-2020-00224 (Bursary for Karabo Tsholo), The Water Research Commission (WRC) of South Africa: (Contract - 2019/2020-00224). The views expressed are those of the authors and not of the funding agencies.

Additional Editor Comments:

Dear authors,

I am pleased to inform you that both reviewers only requested minor revisions for future publication endorsement. Please carefully answer both reviewers' concerns and rectify the manuscript following their comments.

Thank you and best regards,

António Machado

Reviewers' comments:

Reviewer's Responses to Questions

**Comments to the Author**

1. Is the manuscript technically sound, and do the data support the conclusions?

Reviewer #1: Yes

Reviewer #2: Yes

2. Has the statistical analysis been performed appropriately and rigorously? 

Reviewer #1: Yes

Reviewer #2: Yes

3. Have the authors made all data underlying the findings in their manuscript fully available?

Reviewer #1: Yes

Reviewer #2: Yes

4. Is the manuscript presented in an intelligible fashion and written in standard English?

Reviewer #1: Yes

Reviewer #2: Yes

5. Review Comments to the Author

Reviewer #1: The Author has tried o identify the correlation between ARGs detected and the antibiotic residues in the drinking water facilities. However, it is not clear why only few antibiotic resistance genes were selected.

Were bacterial communities identified in the water common inhabitants of drinking water?

It looks like antibiotic residues were selected based on the ARGs identified in the water or vice versa. What is the relevance of this?

Reviewer #2: I consider authors have carried out a valid study. Experiments and analysis are appropiated for answering the research question. Since I have several comments about most of the paper sections (Abstract, Introduction, Methods, Results and Discussion), a review report have been uploaded as a PDF document.

6. PLOS authors have the option to publish the peer review history of their article (what does this mean?). If published, this will include your full peer review and any attached files.

Reviewer #1: No

Reviewer #2: No

---

## [Author Response · Author response to Decision Letter 0]

11 Jan 2024

Comments and Response to comments by authors for Prof José António Baptista Machado Soares

Comment: 1 Please ensure that your manuscript meets PLOS ONE's style requirements, including those for file naming. The PLOS ONE style templates can be found at. Response: Where applicable the style of the manuscripts were revised to match PLOS ONE style templates 

Comment: 2 In your Methods section, please provide additional information regarding the permits you obtained for the work. Please ensure you have included the full name of the authority that approved the field site access and, if no permits were required, a brief statement explaining why. Response: There is a memorandum of understanding between North-West University and the local municipality responsible for the two DWPFs. However, the local municipality granted permission for this study on a condition of anonymity (lines 94 to 95). 

Comment: 3 Please include your amended Funding Statement within your cover letter. We will change the online submission form on your behalf. Response: The Funding Statement has been amended as following: This work is based on the research supported in part by the National Research Foundation of South Africa Grant No. C2019-2020-00224 (Bursary for Karabo Tsholo), The Water Research Commission (WRC) of South Africa: (Contract – C2019-2020-00224). The views expressed are those of the authors and not of the funding agencies. The Funding Statement has been included on the cover letter. 

Comment: 4: We note that the grant information you provided in the ‘Funding Information’ and ‘Financial Disclosure’ sections do not match. Response: The grant information of the Funding Information’ and ‘Financial Disclosure’ now matches. 

Comment: 5 Please include captions for your Supporting Information files at the end of your manuscript, and update any in-text citations to match accordingly. Response: Captions of the Supporting Information files were included at the end of manuscript as following; S1 Table, S2 Table, S3 Table, S1 Appendix. Furthermore, in-text citations were also amended.

 Comments and Response to comments by authors for reviewer 1

Comment:1 The Author has tried to identify the correlation between ARGs detected and the antibiotic residues in the drinking water facilities. However, it is not clear why only a few antibiotic resistance genes were selected. Response: The author tried to correlate bacterial communities with physicochemical parameters, ARGs and antimicrobial residues. Initially, commonly prescribed antimicrobial agents and prevalent ARGs were included in the analysis. However, due to the failed optimization for screening ARGs and quantifying antimicrobial residues, only a few parameters were included in the current study (lines 467 to 471). 

Comment: 2 Were bacterial communities identified in the water common inhabitants of drinking water? Response: In this manuscript, drinking water is referred to as treated water. The bacterial composition of the drinking/treated water can be found in lines 298 to 317. 

Comment: 3 It looks like antibiotic residues were selected based on the ARGs identified in the water or vice versa. What is the relevance of this? Response: The antibiotics and ARGs selected were based on their prescription and prevalence in SA respectively (lines 467 to 471). 

 Comments Response to comments by authors for reviewer 2

Comment: 1 Consider using “antimicrobial agents” instead when it comes to both antibiotics and fluconazole. Response: Thank you for the suggestion to use “antimicrobial agents”. However, the authors felt that the word “Antimicrobials” would fit better when referring to both antibiotics and fluconazole.

Comment: 2 In Introduction justify the importance of monitoring fluconazole or antifungals because antibiotics are not the only antimicrobials quantified in this study. Response: The justification was provided in lines 61 to 64.

Comment: 3 In Methods and materials most of the concentration units are represented as fractions (E.g.,ml/day), but concentration units of flow rate are not. Please represent all of these similarly. Response: The flow rate has been changed from mL min-1 to mL/min for consistency.

Comment: 4 The title of each section is capitalized but is not the same for the Results section. Please use the same format. Response: The format was changed from “results” to “Results”

Comment: 5 Finally, figures should have a good resolution in order to not lose text legibility. Response: Where applicable, images were replaced with higher-quality images. 

Comment: 6 Line 33-34-Please replace “In NW-E, antibiotic and fluconazole concentrations…” with “Regarding antimicrobial agents, antibiotic and fluconazole concentrations. Response: Replaced as recommended

Comment: 7 Line 63-It is appropriate to only use the term “human bodies” instead of “human and human bodies” in the same sentence. Response: It was changed to “human and animal bodies”

Comment: 8 Line 79-Please use “antimicrobial agents ’residues” instead of “antibiotic residues because in the study fluconazole is also quantified. Response: It has been changed to “antimicrobial residues” rather than “antimicrobial agents’ residues” in the suggested line and others as well. 

Comment: 9 Line 95-Please rectify “environmental DNA (eDNA)” Response: It was changed to as recommended “environmental DNA (eDNA)”

Comment: 10 Line 128-Verify the version of QIIME2 used in this study because the version of this software contains the year and the month. E.g., QIIME2 Version 2022.4 Response: The version of QIIME2 was changed to QIIME2 version 2022.2 (line 132)

Comment: 11 Line 142-As a suggestion change “for screening for the presence…” by “for screening the presence…” Response: Changed as recommended (line 146).

Comment: 12 Line 198-The slope symbol is “m”. Please rectify it. Response: The symbol is changed as recommended

Comment: 13 Line 225-226-It is not clarified in which cases student’s t-test and ANOVA were used when data showed a normal distribution. Response: The error was fixed. The student’s t-test was the one used when data showed a normal distribution. ANOVA was deleted.

Comment: 14 Line 274-275-Please detail the temperature results as was made with the other variables. The seasons in which temperature was measured should be specified in Table 2 and Table Response: Since this study was not based on seasonal variation, the statement was changed to “Temperature fluctuated between raw and treated water in NW-E and NW-C. In NW-E, the temperature range was between 10.3˚C and 24.1˚C. In NW-C, the range was between 13.6 ˚C and 20.9˚C. (Tables 2 and 3).” This description matches the ones of other variables.

Comment: 15 Line 299-318-“Unclassified” group should not be mentioned as phyla or genera, so for example in the first case it can be reported that there were six phyla, detailing the proportion from each of them, but a specific percentage of the sequences have not taxonomy assignation. Response: “Unclassified” was removed as part of the phyla and genera

Comment: 16 Line 491-492-Please cite the explanation of nitrite findings. Response: A citation was included [32] Amanatidou E, Samiotis G, Trikoilidou E, Pekridis G, Taousanidis N. Evaluating sedimentation problems in activated sludge treatment plants operating at complete sludge retention time. Water Research. 2015;69:20-9

Comment: 17 Line 679-686-An explanation about the risks of fluconazole presence in drinking water is recommended. Response: The explanation was provided “Its frequent application and recalcitrance to biodegradation in conventional wastewater treatment plants may lead to its prevalence in the water environment which may pose risks to water users”

Comment: 18 Most of the manuscript is well-written in a comprehensive way; nonetheless, it is recommended that redaction could be improved for avoiding redundancy. Response: Reduced were applicable.

---

## [Decision Letter · Decision Letter 1]

8 Feb 2024

Distribution of antibiotic resistance genes and antibiotic residues in drinking water production facilities: links to bacterial community

PONE-D-23-28722R1

Dear authors,

I am pleased to inform you that both reviewers enjoyed the manuscript very much and endorsed the revised manuscript for publication.

Thank you for choosing Plos ONE journal to publish your study.

Best regards,

António Machado

Reviewers' comments:

Reviewer's Responses to Questions

**Comments to the Author**

1. If the authors have adequately addressed your comments raised in a previous round of review and you feel that this manuscript is now acceptable for publication, you may indicate that here to bypass the “Comments to the Author” section, enter your conflict of interest statement in the “Confidential to Editor” section, and submit your "Accept" recommendation.

Reviewer #1: All comments have been addressed

Reviewer #2: (No Response)

2. Is the manuscript technically sound, and do the data support the conclusions?

Reviewer #1: Yes

Reviewer #2: Yes

3. Has the statistical analysis been performed appropriately and rigorously? 

Reviewer #1: N/A

Reviewer #2: Yes

4. Have the authors made all data underlying the findings in their manuscript fully available?

Reviewer #1: Yes

Reviewer #2: Yes

5. Is the manuscript presented in an intelligible fashion and written in standard English?

Reviewer #1: Yes

Reviewer #2: Yes

6. Review Comments to the Author

Reviewer #1: The manuscript can now be accepted for publication, since the authors have meticulously followed all the suggestion from the reviewers

Reviewer #2: (No Response)

7. PLOS authors have the option to publish the peer review history of their article (what does this mean?). If published, this will include your full peer review and any attached files.

Reviewer #1: No

Reviewer #2: No

---

## [Editor Report · Acceptance letter]

27 Mar 2024

PONE-D-23-28722R1 

PLOS ONE

Dear Dr. Tsholo, 

I'm pleased to inform you that your manuscript has been deemed suitable for publication in PLOS ONE. Congratulations! Your manuscript is now being handed over to our production team.

Kind regards, 

on behalf of

Dr. António Machado 

Academic Editor

PLOS ONE